# *Trypanosoma cruzi* STIB980: A TcI Strain for Drug Discovery and Reverse Genetics

**DOI:** 10.3390/pathogens12101217

**Published:** 2023-10-04

**Authors:** Anna Fesser, Sabina Beilstein, Marcel Kaiser, Remo S. Schmidt, Pascal Mäser

**Affiliations:** 1Swiss Tropical and Public Health Institute, Department Medical Parasitology and Infection Biology, 4123 Allschwil, Switzerland; anna.fesser@posteo.de (A.F.); sabina.beilstein@swisstph.ch (S.B.); marcel.kaiser@swisstph.ch (M.K.); remo.schmidt@agroscope.admin.ch (R.S.S.); 2University of Basel, 4001 Basel, Switzerland

**Keywords:** *Trypanosoma cruzi*, genome sequencing, reverse genetics, drug efficacy testing

## Abstract

Since the first published genome sequence of *Trypanosoma cruzi* in 2005, there have been tremendous technological advances in genomics, reverse genetics, and assay development for this elusive pathogen. However, there is still an unmet need for new and better drugs to treat Chagas disease. Here, we introduce a *T. cruzi* assay strain that is useful for drug research and basic studies of host–pathogen interactions. *T. cruzi* STIB980 is a strain of discrete typing unit TcI that grows well in culture as axenic epimastigotes or intracellular amastigotes. We evaluated the optimal parameters for genetic transfection and constructed derivatives of *T. cruzi* STIB980 that express reporter genes for fluorescence- or bioluminescence-based drug efficacy testing, as well as a Cas9-expressing line for CRISPR/Cas9-mediated gene editing. The genome of *T. cruzi* STIB980 was sequenced by combining short-read Illumina with long-read Oxford Nanopore technologies. The latter served as the primary assembly and the former to correct mistakes. This resulted in a high-quality nuclear haplotype assembly of 28 Mb in 400 contigs, containing 10,043 open-reading frames with a median length of 1077 bp. We believe that *T. cruzi* STIB980 is a useful addition to the antichagasic toolbox and propose that it can serve as a DTU TcI reference strain for drug efficacy testing.

## 1. Introduction

Chagas disease is a neglected tropical and most elusive disease [1,2]. Given the chronic nature of Chagas disease, with an indeterminate phase that is asymptomatic and lasts for decades, the vast majority of the carriers do not know that they are infected. For the same reason, there are no solid data on the prevalence of Chagas disease. The epidemiology of Chagas disease is further complicated by (i) the large zoonotic reservoir of *Trypanosoma cruzi*, which infects all kinds of mammals provided they are preyed upon by the triatomine vectors [3]; (ii) alternative transmission routes, including via the oral mucosa upon consumption of contaminated food [4], via blood or organ donation [5], and transplacental to an unborn child [6]; (iii) the genetic heterogeneity and genomic flexibility of *T. cruzi* with its (at least) seven different discrete typing units (DTUs) [7,8].

These parasites are also elusive in the human body. *Trypanosoma cruzi* can infect any type of nucleated cell, and the parasites will replicate intracellularly in the cytosol of the host cell. Infected macrophages distribute the parasites throughout the body. Thus, they can access different tissues and niches to hide in, including the heart and the intestinal tract, the typical sites of chronic pathology [9,10]. Trypomastigote *T. cruzi* do not proliferate but persist extracellularly in the blood thanks to their elaborate immune evasion strategies [11]. The intracellular amastigotes, too, can enter a non-replicative state of dormancy [12,13]. All this makes Chagas disease difficult to diagnose and even harder to cure, as became apparent in clinical trials with new antichagasic drug candidates [14,15]. In the laboratory, research on *T. cruzi* is hampered by the fact that the disease-relevant stages, the amastigotes, are strictly intracellular and require host cells for in vitro culture. The infectious nature of *T. cruzi* renders all experimental investigation resource-intensive in terms of biosafety measures, assay time, and overall cost [16].

On a positive note, there has been tremendous technological progress in genomics and reverse genetics with *T. cruzi*, which has boosted basic research and drug discovery. Classical genetic manipulation based on homologous recombination [17,18] is being replaced by CRISPR/Cas9-mediated gene editing [19], which allows for functional genomics in spite of the fact that *T. cruzi* lacks RNA interference machinery [20]. Genetically engineered reporter strains of *T. cruzi* have enabled assay formats that better predict the potential of antichagasic molecules for irreversible and cidal action, both in vitro and in vivo [21,22]. Here, we present the reference strain *T. cruzi* STIB980, which is useful for all kinds of investigations including genomics, reverse genetics, and drug efficacy testing.

## 2. Materials and Methods

### 2.1. Cells and Cultivation

*T. cruzi* STIB980 was originally received in 1983 from Prof. Antonio Osuna, University of Granada. Epimastigotes were cultured at 27 °C in LIT medium supplemented with 2 µg/mL hemin and 10% heat-inactivated fetal calf serum (iFCS) [23]. The cultures were diluted weekly. Metacyclogenesis was stimulated by keeping the epimastigotes for 3 to 4 weeks in the same medium. Mouse embryonic fibroblasts (MEFs) were cultured at 37 °C, 5% CO_2_ in RPMI medium supplemented with 10% iFCS and >95% humidity. The MEFs were subpassaged weekly at a ratio of 1:10 after 5 min treatment with trypsin. Peritoneal mouse macrophages (PMMs) were obtained from female CD1 mice. A 2% starch solution in distilled water was injected i.p., and the macrophages were harvested 24 h later via peritoneal lavage. The cells were washed and resuspended in RPMI medium containing 1× antibiotic cocktail [24], 10% iFCS, and 15% medium conditioned by LADMAC cells (ATCC^®^ CRL2420™), which secrete colony-stimulating factor 1 (CSF-1). The macrophages were kept in this medium at 37 °C for 3 to 4 days and then detached with trypsin treatment and cell scrapers. The isolation of PMMs from mice was conducted in accordance with the strict guidelines set out by the Swiss Federal Veterinary Office, under the ethical approval of license number #2374.

### 2.2. Cloning of T. cruzi

The gilded paper clip method was used for cloning (Figure 1A). An exponentially growing epimastigote culture was diluted to 5 × 10^4^ cells/mL. The outer wells of a 96-well plate were filled with 100 μL sterile water. 15 μL of conditioned LIT medium supplemented with 10% filtered post-culture medium and 20% iFCS was placed at the edge of the other wells so that some space on the well remained dry. Using a gold-plated paperclip, a micro-drop of approximately 0.1 μL was transferred from the diluted parasite suspension to the dry space of the well. Two people analyzed the droplet under an inverted microscope. Wells that contained only one parasite were supplemented with 35 μL of conditioned LIT. The plates were incubated at 27 °C and assessed regularly for the outgrowth of the clones.

### 2.3. Isolation of Genomic DNA

Genomic DNA for genome sequencing was isolated from 10^8^ epimastigotes. The cells were washed and resuspended in 500 µL of NTE and lysed via the addition of 25 µL of 10% SDS. The lysate was treated with 50 µL of RNase A (10 mg/mL) and 25 µL of pronase (20 mg/mL) and incubated overnight at 37 °C. The lysate was extracted sequentially with phenol and chloroform:isoamyl alcohol (24:1). The DNA was precipitated via the addition of 1 mL of cold absolute ethanol. For Illumina sequencing, the DNA was pelleted via centrifugation; for Oxford Nanopore sequencing, the DNA was collected with a glass hook. The DNA was washed with 70% ethanol, air-dried, and resuspended in 80 µL of DNase-free water. For other purposes, genomic DNA was isolated with the QIAGEN DNeasy blood and tissue kit. DNA quality control was performed with Nanodrop (Mettler Toledo, Columbus, OH, USA), using OD_260_/OD_280_ >1.8 and OD_260_/OD_230_ between 2.0 and 2.2 as inclusion criteria and quantified fluorometrically with QuBit (Thermo Fisher, Waltham, MA, USA). For Illumina sequencing, the genomic DNA was fragmented via sonication to a medium insert size of 750 bp. For Oxford Nanopore sequencing, the genomic DNA was used as isolated.

### 2.4. Genome Sequencing and Assembly

Library preparation and sequencing on the Illumina platform were performed at the Quantitative Genomics Facility Basel (GFB) of the ETH Zürich. Sequencing libraries were prepared using the PCR-free KAPA HyperPrep kit (Illumina, San Diego, CA, USA). Paired-end sequencing of 125 nucleotides was performed with an Illumina HiSeq 2500 sequencer. For Nanopore sequencing, the library was prepared using the Ligation Sequencing kit 108 (SQK-LSK108, Oxford Nanopore Technology, Oxford, UK) and sequenced using the MinION (1D R9.3) platform. Basecalling was carried out using Albacore. Quality control for all reads was performed with FastQC (version 0.11.3) [25]. The reads were trimmed stringently using the following Trimmomatic [26] parameters: SLIDINGWINDOW:4:30, LEADING:10, TRAILING:10, HEADCROP:6, and MINLEN:36. This left 38.60% of paired reads and an additional 23.71% and 5.18% of forward- and reverse-only reads, respectively. Thus, 32.51% of the original 67,187,531 read pairs were discarded. We benchmarked different assemblers available at the time: Velvet [27] and SOAPdenovo2 (version 2.04) [28] for the Illumina reads (with a range of different kmer sizes, from 17 to 73), Canu (version 1.7) [29], and Flye (release 2.3.3) [30] for the Nanopore reads. Velvet and SOAPdenovo2 assembled the genome in the lowest amount of contigs. For Velvet, we followed the tutorial by Thomas Otto [31]. At kmer size 55, we had the best results in terms of contig number and N50. On this assembly, 95.07% of the trimmomatic-filtered single reads were mapped, as were 73.49% of the paired reads. The best mapping result with SOAPdenovo2 was at kmer size 17, with 79.66% and 32.34% mapping for single and paired reads, respectively. Illumina polishing of the Canu-assembled Nanopore reads was performed using Pilon (version 1.22) [32], followed by BWA-MEM [33] with default parameters. The Flye assembly was performed on the pore-chopped long reads, with an expected genome size of 53 Mb [30]. Gene prediction was performed using GLIMMER [34] with the standard codon table. The genome of *T. cruzi* Dm28c [35] served as the training set.

### 2.5. Optimization of Electroporation

10^7^ epimastigotes from a dense culture were centrifuged and resuspended in 100 µL of TbBSF buffer [36] containing 10 µg of circular (for transient transfection) or linearized (for stable transfection) plasmid DNA. The plasmid pTcRG was kindly provided by Santuza Teixeira (Federal University of Minas Gerais, Belo Horizonte, Brazil). The cells were electroporated with a nucleofector device (Lonza) in a 0.2 mm cuvette (BioRad, Hercules, CA, USA). After electroporation, the cells were transferred to 10 mL of LIT with a fine-tipped Pasteur pipette. The parasites transfected with circular plasmids were incubated for 24 h and then tested for GFP expression with flow cytometry with a FACSCalibur machine (Becton Dickinson and Company, Franklin Lakes, NJ, USA). The parasites transfected with linearized plasmid were incubated for 24 h, diluted 1:10 in medium containing 100 µg/mL G418 (Gibco, Billings, MT, USA), and further distributed in a fourfold dilution series in a 48-well plate under antibiotic pressure. Outgrowing epimastigotes were cloned by limiting dilution and assessed for correct integration of the transgene with PCR and Southern blot.

### 2.6. Generation of Transgenic Lines

Genetic transfection and CRISPR-Cas9-mediated genetic knockout were performed according to [37,38]. In brief, exponentially growing *T. cruzi* epimastigotes were synchronized for 24 h with 20 mM of hydroxyurea (Sigma, Burlington, MA, USA) [39]. Following hydroxyurea removal via washing twice with PBS, 10^7^ epimastigotes were electroporated with 2.5 μg of pTRIX2 Luc::Neon-HYG plasmid linearized with *Asc*I and *Sac*I (New England Biolabs) or pLEW-Cas9 plasmid linearized with *Not*I (New England Biolabs, Ipswich, MA, USA). The pTRIX2 Luc::Neon-HYG plasmids were derived from pTRIX-REh9 [40]. The plasmids were kindly provided by John Kelly (LSHTM). The resistance cassette was amplified via PCR from plasmid pPOTcruzi v1 blast-blast mNeonGreen with primers 5’ aacatcaagaagggaccagcccccttctacggtagttaagagctcggacccac (forward) and 5’ acgagtgctggggcgtcggtttccactatcccaatttgagagacctgtgc (reverse). sgRNA were amplified using a gene-specific forward primer (5’ gaaattaatacgactcactatagggagtacttctacacagccatgttttagagctagaaatagc and 5’ gaaattaatacgactcactataggcggctgtgccgtcctccagggttttagagctagaaatagc) and the G00 (sgRNA scaffold) reverse primer. Twenty-four hours after transfection, the parasites were diluted 1:10 in medium containing 100 μg/mL G418 (Gibco).

### 2.7. Drug Sensitivity Assay with Epimastigotes

In a 96-well microtiter plate, 100 µL of epimastigotes at a starting density of 5 × 10^6^/mL, 10^5^/mL, or 2 × 10^4^/mL was incubated with a test compound in threefold serial dilution with 11 dilution steps. After 69 h or 165 h of incubation at 27 °C, 10 µL of resazurin (Sigma) solution (12.5 mg in 100 mL water) was added to each well. After another 3 h of incubation, the plates were read with a SpectraMAX GeminiXS fluorescence reader (Molecular Devices, San Jose, CA, USA), and 50% inhibitory values (IC50) were determined in R version 3.5.1 (R Core Team 2018, Vienna, Austria) using the “drc” package [41].

### 2.8. Flow Cytometry

For flow cytometry, 10^5^ epimastigotes were fixed with 10% formalin (Sigma) for 15 min and then analyzed for their green fluorescence levels (FL1) with a BD FACSCalibur (Becton Dickinson and Company, Franklin Lakes, NJ, USA). The threshold for GFP expression was set above the autofluorescence level of 99.6% of the untransfected control cells. The proportion of GFP-expressing cells was defined as the proportion of cells exhibiting a higher level of fluorescence than the threshold.

### 2.9. High-Content Drug Efficacy Assay

Assays were performed with two technical and two biological replicates. For the standard assay, 10^4^ PMMs were seeded into the central wells of a black 96-well plate (Greiner, uClear, black, REF 655090, Lot E1803364) in 100 µL of RPMI medium containing 1% antibiotic mix [24], 10% iFCS, and 15% RPMI containing LADMAC growth factors per well. The border wells were filled with 100 µL of water. After 48 h, the PMMs were infected with 10^4^ trypomastigotes from either the wildtype or the transgenic STIB980 line. After 24 h, the remaining trypomastigotes were washed off twice with 200 µL of RPMI per well. The infected PMMs were kept in 100 µL RPMI containing 1% antibiotic mix and 10% iFCS. Drugs were added in threefold serial dilutions 24 h post-infection. At 96 h after the addition of drugs, the plates were fixed with 10% formalin for 15 min at room temperature. Subsequently, the plates were stained with 50 µL of 5 µM Draq5 (BioStatus, Leicester, UK) per well for 30 min at room temperature in the dark. The plates were stored at 4 °C for at least 24 h and then imaged using an ImageXpress Micro XLS microscope (Molecular Devices, San Jose, CA, USA) with a 20× Zeiss objective with a Cy5 filter cube for 300 ms per image on 9 sites per well. Image analysis was performed with the MetaXpress 6 software. Statistical analysis and graphs were performed in R version 3.5.1 (R Core Team 2018) using the packages “tidyverse” [42] and “readxl” [43].

## 3. Results and Discussion

### 3.1. Genotyping and Cloning of T. cruzi STIB980

*T. cruzi* STIB980 is one of the standard strains used for drug efficacy testing at the Parasite Chemotherapy Unit of the Swiss TPH. Amastigote and epimastigote forms are readily cultured as described in the Methods section. A fresh clone of *T. cruzi* STIB980 was made with epimastigotes, employing the gilded paperclip method (Figure 1A).

This clone of *T. cruzi* STIB980 was used for all further analyses. Genotyping based on the restriction fragment length polymorphisms of three target loci (the large ribosomal RNA subunit, heat-shock protein 60, and glucose-6-phosphate isomerase (PGI)) [44] placed *T. cruzi* STIB980 in DTU TcI (Figure 1B–F). This was confirmed constructing a phylogenetic tree of the *PGI* nucleotide sequences, in which STIB980 clustered with the DTU TcI branch (Figure 2). TcI is one of the DTUs that circulates most broadly among humans, and it correlates with cardiomyopathy [45,46]. Therefore, a TcI strain is highly relevant as an assay strain.

### 3.2. Genome Sequence of T. cruzi STIB980

The genomic DNA of *T. cruzi* STIB980 was sequenced with the Illumina and Oxford Nanopore technologies. Illumina sequencing was performed with a 125 bp paired-end protocol and yielded 45,345,000 reads that passed quality control. With Nanopore sequencing, we obtained 250,005 reads and a median length of 1.4 kb (Figure 3A). Taking the actual read lengths and assuming a haploid genome size of 53.3 Mb, as reported for *T. cruzi* Dm28c [35], this provides a total coverage of 25.3-fold for the Nanopore sequencing alone. The coverage of the nuclear genome, i.e., excluding mini- and maxicircles, was 19.4-fold. The reads were categorized according to their size and GC content (Figure 3) into nuclear genome, maxicircle (assembled to a single contig and confirmed with blastn searches), minicircles, and sequences of unknown origin (Table 1).

The best results for genome assembly (as judged by the mapping rate) were obtained by first assembling the long Nanopore reads (using Canu v1.7 [29]), followed by fixing errors with the short Illumina reads (using Pilon v1.22 [32]). This combination of Nanopore and Illumina reads led to drastic improvements compared with the assembly based on Illumina reads alone: the number of contigs was reduced 23-fold, the N50 increased 30-fold, and the number of gaps (n = 13,000) and undetermined nucleotides (5 Mb) were reduced to zero. The total assembly amounted to 28.2 Mb in 492 contigs (Figure 3B); the nuclear genome had a haploid size of 27.9 Mb in 397 contigs (Table 1). This is at the lower end of the range of published *T. cruzi* genome sizes, which vary from 27 Mb to 83 Mb [53].

The gene prediction was based on *T. cruzi* Dm28c [35] as the training set, and it resulted in 10,043 open-reading frames (ORFs) with a median length of 1077 bp. The amino acid sequences were queried against the UniProt KnowledgeBase [54] using blastp [50] with an expectancy (E-value) cut-off of 10^−8^. This allowed for the functional annotation of 3505 genes.

### 3.3. Antibiotic Sensitivity Profile of Epimastigote T. cruzi STIB980

In order to determine the best selection markers for use in genetic manipulation, we tested the sensitivity of *T. cruzi* STIB980 epimastigotes to commonly used antibiotics: blasticidin, G418, hygromycin, phleomycin, and puromycin. Benznidazole and nifurtimox were included as benchmark drugs, and DMSO was included as the most commonly used solvent of test compounds. Drug sensitivity was tested for 72 h and 168 h of incubation. For the latter, we used two different inocula: a lower starting density (2 × 10^4^ epimastigotes/mL) to assess the inhibition of proliferation and a higher density (10^5^ epimastigotes/mL) to measure cidality. However, the obtained IC_50_ values were similar across all the tested conditions (Table 2). The STIB980 epimastigotes had comparably high IC_50_ values for G418, which is in agreement with the high concentrations (100 to 500 µg/mL) of G418 that are generally used for epimastigote *T. cruzi* [39] and in stark contrast to the 1 to 5 µg/mL used in the genetic manipulation of procyclic *T. brucei* [36].

Besides the sensitivity of the untransfected trypanosomes, other factors will determine the optimal concentration of antibiotics for selecting positive transfectants. The expression level of the resistance gene will be affected by its copy number (especially in episomal transfections) and the strength of the promoter, the RNA polymerase (RNAPolII, usually resulting in a lower level of transcription than RNAPolI), and—in the case of the ribosomal locus—the exact site of integration [55]. Overall, we recommend blasticidin or puromycin to select for *T. cruzi* STIB980 transfectants rather than G418, hygromycin, or phleomycin.

### 3.4. Optimal Transfection Protocol for T. cruzi STIB980

Lonza nucleofector 2b is a widely used electroporation device for genetic transfection. It also provides excellent results with trypanosomes but is a black box, as the provider does not disclose the characteristics of the electric discharge nor the composition of the buffers. Tests on nucleofector programs have already been published for *T. brucei* [36] and *T. cruzi* [56]. We investigated which program is best suited for *T. cruzi* STIB980. Epimastigotes were transfected with a circular pTcRG plasmid that contained the green fluorescent protein (GFP) gene plus the 3’ UTR of the GAPDH gene, which confers constitutive expression. In total, 4 × 10^7^ epimastigotes in the exponential growth phase were transfected with 10 µg of plasmid DNA using nine different nucleofector programs. Immediately after transfection, we counted the surviving parasites. Then, we incubated them for 24 h in 10 mL of LIT medium at 27 °C. Finally, the proportion of GFP-expressing parasites was quantified via flow cytometry [36]. The transfection efficiency was calculated as the product of cell survival and GFP positivity (Table 3). The programs U-033, X-001, and Z-001 had the best overall efficiencies. The lower survival rates with Z-001 and U-033 were compensated by higher fractions of GFP expression. Program X-001 was recommended for the transfection of *Leishmania mexicana* promastigotes [57]. For subsequent transfections, we used the nucleofector programs U-033 or X-014 [56].

### 3.5. Transgenic T. cruzi STIB980 Lines for Drug Testing and Reverse Genetics

The levels of cytosolic GFP obtained after the stable transfection of pTcRG were too low for high-content fluorescence microscopy. Most of the parasite signal was below three times the background level (i.e., the autofluorescence of untransfected epimastigotes). For better use of *T. cruzi* STIB980 in drug efficacy testing and molecular genetics, we generated stable transgenic lines expressing a *LucNeon* reporter gene, a *Cas9* nuclease gene, or both [37]. *LucNeon* is a chimeric gene that encodes a fusion protein of mNeonGreen, suitable for fluorescence-based in vitro imaging, plus a red-shifted luciferase that is suitable for bioluminescence-based in vivo imaging [37]. Epimastigotes were transfected as described in the Methods section. The three resulting transgenic lines all had similar growth rates with population-doubling times around 20 h, slightly higher than the 17 h of the parental *T. cruzi* STIB980 (Figure 4).

The sensitivity profiles to reference drugs (benznidazole and nifurtimox) and drug candidates (posaconazole, fexinidazole, and oxaborole DNDi-6148) of parental *T. cruzi* STIB980 and STIB980-LucNeon were determined using high-content imaging of intracellular amastigotes in expanded mouse peritoneal macrophages. The IC_50_ values were calculated with two different methods, either based on the number of infected host cells or the total number of intracellular amastigotes (Table 4).

The first method resulted in slightly higher IC_50_ values, which was to be expected as the total number of parasites can be reduced more readily than host cells cured of the infection. Overall, the drug sensitivities of the parental STIB980 and transgenic derivative were very similar using both methods (Table 4). The function of the Cas9 nuclease was validated via the CRISPR/Cas9-mediated deletion of the fluorescence reporter using specific guide RNA for the *LucNeon* gene (Figure 5).

## 4. Conclusions

*Trypanosoma cruzi* STIB980 is a useful new assay strain in the toolbox of antichagasic drug discovery. It is a DTU TcI strain that is readily cultured in vitro and amenable to genetic manipulation. We provide optimized electroporation conditions and the antibiotic sensitivity profile of epimastigotes to facilitate genetic transfection. The genome sequence of *T. cruzi* STIB980 was assembled by combining short reads generated with Illumina sequencing and long reads generated with Oxford Nanopore sequencing, demonstrating the power of combining both technologies, in particular for a genome with a high degree of repetitive regions like that of *T. cruzi*. We further provide *T. cruzi* STIB980 derivatives that express reporter genes (eGFP, LucNeon) for imaging in vitro and in vivo. The reporter genes are stable in the absence of selective pressure in epimastigotes but much less so in amastigotes, underlining the importance of frequently resorting to a new stabilate, e.g., when running drug-testing campaigns against intracellular amastigotes. To facilitate CRISPR/Cas9-mediated gene editing, we also constructed a line of *T. cruzi* STIB980-LucNeon with a stably integrated *Cas9* gene and validated that line by knocking out the *LucNeon* gene as a proof-of-principle. Thus, *T. cruzi* STIB980 can serve not only as a reference strain for drug efficacy testing but also as a tool for molecular genetics.

## Figures and Tables

**Figure 1 pathogens-12-01217-f001:**
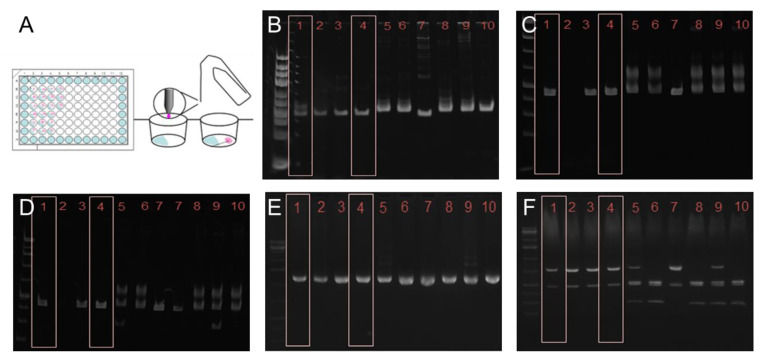
Establishing a *T. cruzi* STIB980 clonal line with the gilded paperclip method (**A**) and genotyping results. Agarose gels of the PCR products of the large ribosomal subunit (**B**); HSP60 before (**C**) and after (**D**) digestion with *Eco*RV; PGI before (**E**) and after (**F**) digestion with *Hha*I. In all three reactions and subsequent digestions, STIB980 most closely resembled the DTU TcI strains. (1) Dm28c (TcI); (2) Sylvio (TcI); (3,4) STIB980; (5) Tulahuen (TcVI); (6) Esmeraldo (TcII); (7) Sylvio X10/4 (TcI); (8) Y strain (TcII); (9) CL Brener (TcVI); (10) Y strain (TcII). Genomic DNA was kindly provided by Michael Lewis (LSHTM).

**Figure 2 pathogens-12-01217-f002:**
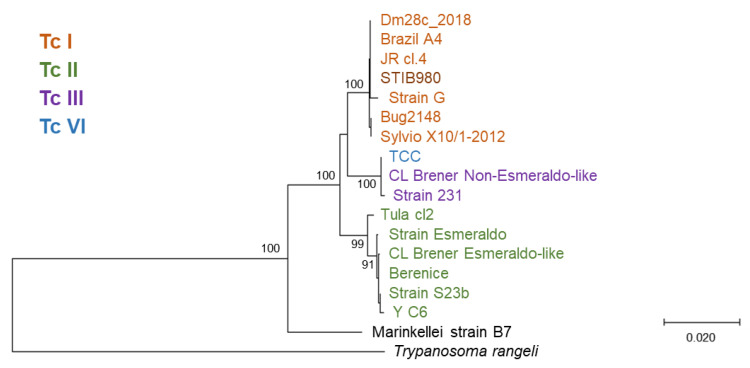
Neighbor-Joining phylogenetic tree of PGI coding sequences. The naming of the *T. cruzi* strains is that of TriTrypDB [47]; discrete typing units are color-labeled [48,49]. *T. cruzi marinkellei* and *T. rangeli* are included as outgroups. All nucleotide sequences were downloaded from tritrypdb.org after a blastn [50] search with STIB980 *PGI* as the query sequence. Multiple alignment was performed with MUSCLE [51] using default parameters, and the tree was drawn with MegaX [52]. Bootstrap values are percent positives of 1000 rounds; only values above 90 are shown. The scale bar indicates the number of base substitutions per site.

**Figure 3 pathogens-12-01217-f003:**
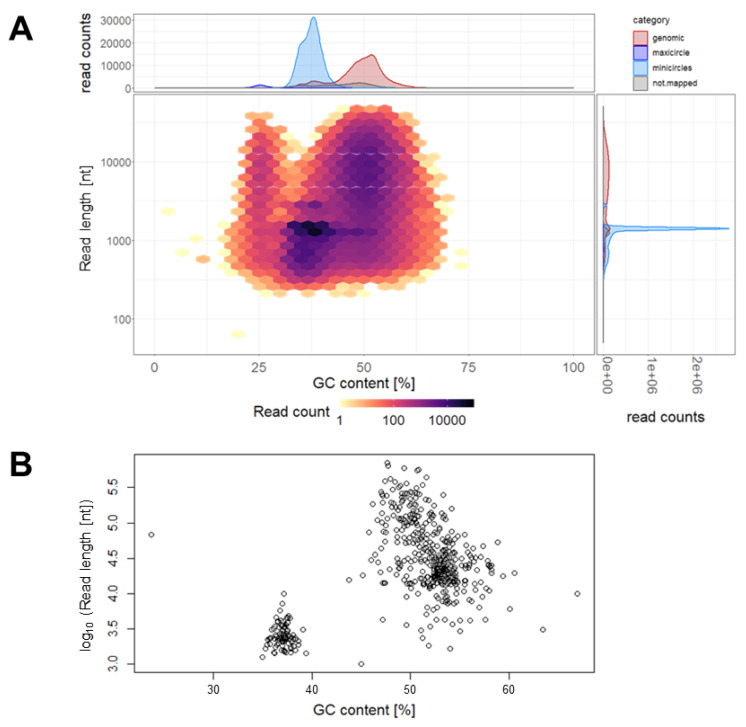
Distribution of the Nanopore reads ((**A**), n = 250,005) and assembled contigs ((**B**), n = 492) of *T. cruzi* STIB980 according to their GC content and length. This separates nuclear sequences from mitochondrial sequences. The majority of the reads were categorized as minicircles, with GC content between 30% and 40% and a length of about 1.4 kb.

**Figure 4 pathogens-12-01217-f004:**
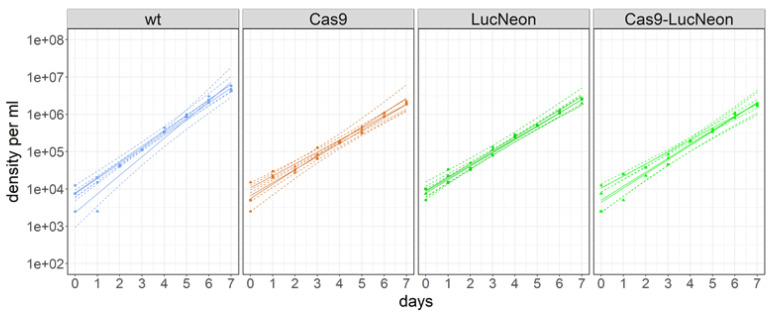
Growth curves of epimastigote *T. cruzi* STIB980 wildtype (wt, blue) and transgenic derivative-expressing Cas9 nuclease (orange), *LucNeon* reporter gene (green), or both (bright green). The indicated population doubling times were calculated via linear regression to the log-transformed data.

**Figure 5 pathogens-12-01217-f005:**
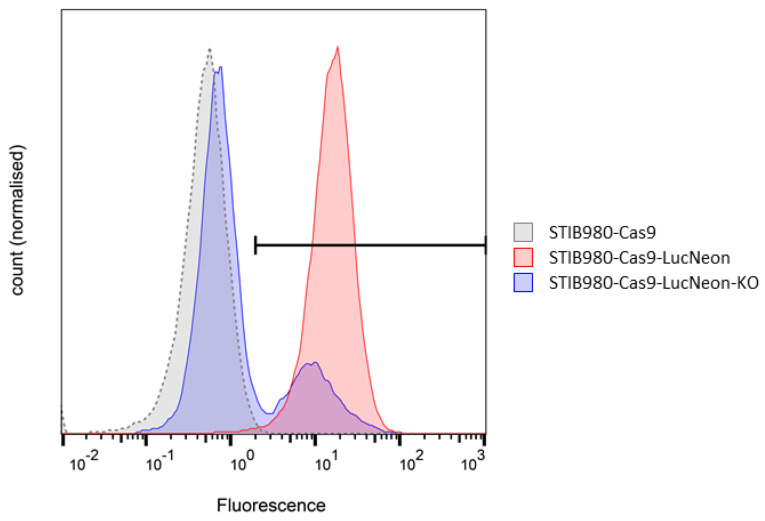
Validation of the LucNeon reporter and Cas9 nuclease in *T. cruzi* STIB980-Cas9-LucNeon via flow cytometry. The x-axis represents the fluorescence level in arbitrary units, measured with the green fluorescence channel (excitation, 488 nm; emission, 525 nm; bandwidth, 50 nm); the y-axis is the normalized cell count. Only 23.7% of the cells still showed a green fluorescence signal after CRISPR-Cas9-mediated knockout of the *LucNeon* fusion gene.

**Table 1 pathogens-12-01217-t001:** Summary statistics of the separate genome assemblies for *T. cruzi* STIB980.

	Total Size (nt)	No. of Contigs	N50 (nt)	Longest Contig (nt)	Shortest Contig (nt)
Nuclear	27,888,483	397	165,577	715,804	1660
Minicircles	248,782	91	2699	10,035	1264
Maxicircle	68,708	1	n.a.	n.a.	n.a.
Unknown	14,070	3	3090	9979	1001
Total	28,220,043	492	158,042	715,804	1001

**Table 2 pathogens-12-01217-t002:** Antibiotic sensitivity profile of *T. cruzi* STIB980 epimastigotes. All values are µg/mL; the 95% CIs are provided in parentheses.

	IC_50_ 72 h High Inoculum ^1^	IC_50_ 168 h High Inoculum ^2^	IC_50_ 168 h Low Inoculum ^3^
Blasticidin	1.6 (1.2; 2.0)	0.37 (0.20; 0.54)	0.32 (0.29; 0.34)
Puromycin	1.3 (1.1; 1.4)	1.2 (1.1; 1.4)	0.56 (0.48; 0.64)
Hygromycin	41 (30; 52)	22 (13; 31)	37 (28; 46)
G418	46 (38; 54)	50 (42; 57)	31 (25; 37)
Phleomycin	89 (70; 110)	71 (60; 81)	27 (23; 32)
Benznidazole	1.9 (0.67; 3.1)	1.2 (0.90; 1.4)	0.66 (0.53; 0.79)
Nifurtimox	0.87 (0.58; 1.2)	0.41 (0.37; 0.46)	0.24 (0.20; 0.28)
DMSO	3.8 (3.2; 4.5)	3.2 (−1.1; 7.4)	1.2 (0.93; 1.4)

^1^ 5 × 10^5^ mL^−1^; ^2^ 1 × 10^5^ mL^−1^; ^3^ 2 × 10^4^ mL^−1^.

**Table 3 pathogens-12-01217-t003:** Efficiency of transient transfection of different nucleofector programs, expressed as the fraction of surviving cells multiplied by the fraction of GFP-expressing cells.

Program	% Survival	% GFP Expression	Efficiency
X-001	55.0	6.83	0.038
U-033	42.5	8.15	0.035
Z-001	32.5	7.80	0.025
X-014	58.8	4.16	0.024
X-013	57.5	4.23	0.024
X-024	65.0	3.56	0.023
Z-014	60.0	3.18	0.019
X-006	57.5	2.67	0.015
T-020	91.3	1.11	0.010

**Table 4 pathogens-12-01217-t004:** Drug sensitivity profiles of *T. cruzi* STIB980 wildtype (wt) and STIB980-LucNeon as determined using high-content imaging of intracellular amastigotes. IC_50_ values were calculated based on the number of infected host cells (infection rate, left) or the total number of intracellular amastigotes (no. of amastigotes, right).

	IC_50_ (ng/mL) Calculated from Infection Rate		IC_50_ (ng/mL) Calculated from No. Amastigotes	
	wt	LucNeon	wt	LucNeon
Benznidazole	750	840	550	600
Fexinidazole	3000	4500	2900	2300
DNDi-6148	110	110	100	70
Nifurtimox	580	700	<410	540
Posaconazole	0.85	0.72	0.51	0.69

## Data Availability

All sequencing reads were deposited at the National Center for Biotechnology Information as BioProject PRJNA1009388. The assay strain *T. cruzi* STIB980 and its transgenic derivatives described herein are available from the authors.

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
