# Peer review of "Trypanosoma cruzi STIB980: A TcI Strain for Drug Discovery and Reverse Genetics"

_pathogens, 2023, doi:10.3390/pathogens12101217_

Round 1

Reviewer 1 Report

The autors report a novel laboratory strain of T. cruzi DTU I, genetically engineered to be used in screening of compounds with potential tripanocidal activity. The parasites could be used in tests both in vitro and in vivo, but also in molecular genetic research owing to the incorporation of a CRISP/Cas9 site. The new strain developed by the authors represents an improvement in the pool of parasites available for research in the field of Chagas disease, especially contributing with the less represented DTU I. There are only 2 minor comments, one is that someone who is infected with T. cruzi but does not know it would be a patient and not a “carrier” (line 30). The other is that I couldn’t find ref. [35] in the text, although it is listed in the references.

Author Response

We thank the Reviewer for the encouraging comments.

  1. Someone who is infected with T. cruzi but does not know it would be a patient and not a “carrier” (line 30).

To our understanding, someone who is asymptomatically infected is not a patient but a carrier. So we prefer to leave the sentence as is.

  1. I couldn’t find ref. [35] in the text, although it is listed in the references.

Ref. 35 is cited in line 143 (of the revised ms).

Reviewer 2 Report

The manuscript by  Fesser and coworkers focuses on the STIB980 strain of Trypanosoma cruzi, mainly in the context of drug discovery for Chagas Disease. The presented strain shows promise for drug research and basic studies in host-pathogen interaction. Efforts have been made to optimize parameters for genetic transfection, and variants that express reporter genes for drug efficacy testing and a Cas9-expressing line for CRISPR/Cas9-mediated gene editing were generated. In addition, the genome of the strain has been sequenced by using Nanopore and Illumina technologies. While the work is interesting, there are several significant issues that should be considered. They would allow a more comprehensive understanding and evaluation of the findings.

Firstly, methods about genome sequencing and assembly are poorly described. It is recommended to provide a detailed description for the preparation of the sequencing libraries. This should include quality control measures for the DNA, assessments of DNA fragmentation, sequencing details such as the nanopore flow cell version, among other relevant details. In addition, nor quality controls of the reads are mentioned and many software are mentioned for assembling but there is no detail about how they were implemented (e.g. which parameters were set).

Secondly, the genome analysis results are presented superficially. Merely knowing the GC content and mapping frequencies to various sequence types, such as minicircles, maxicircles, and nuclear DNA, doesn't provide a comprehensive understanding of the genome's significance and quality. A comparison with other available Trypanosoma cruzi genomes, like Dm28c which is also TcI, would be insightful.

Thirdly, the typing using PCR-RFLP seems a bit “old” given the availability of the complete genome sequencing. A more robust typing approach would be to conduct a phylogenetic analysis using as many genomic regions as possible. This analysis should encompass both nuclear and mitochondrial levels, keeping in mind that sometimes both might be discordant. Alternatively, extracting reads from gene fragments (e.g. those used in nuclear and maxicircle MLST schemes) and comparing them with a broad panel of strains can provide insights into the DTU to which it belongs.

Lastly, manuscript should be better organized because the Results and Discussion sections have several paragraphs that should be better suited to Material and Methods (See below).

Detailed comments

Abstract:

Line 14: Please use ‘T. cruzi’ instead ‘Trypanosoma cruzi’ when the full species name was already mentioned.

Introduction:

Line 28: The sentence "Chagas disease is a neglected tropical disease and at the same time also a most elusive disease." can be rephrased to avoid redundancy (3x the word ‘disease’).

Lines 59-60. “Here we present a new reference strain, T. cruzi STIB980…” Is this strain a new strain? Which are the differences between this strain and the STIB980 presented in reference [21] ( PLoS Negl Trop Dis 2020, 14, e0008487). In addition, this sentence is contradictory to the sentence in line 177 “Trypanosoma cruzi STIB980, originally received in 1983 from Prof. Antonio Osuna, University of Granada…”

Materials and methods:

Line 104. Which Flow cell did you use?

Line 107. There is no detail about quality control. Which were the criteria? Did you discard low quality reads?

Line 108-109. Did you trimmed Illumina sequences before assembling? Why did you selected such Assemblers (please justify). Which parameters did you used? Which results did you obtained from different kmer sizes?

Lines 110-111. How did you mapped illumina reads to Nanopore reads to use Pilon? Which software did you used? Which parameters?

Lines 133-134. “CRISPR-Cas9 mediated genetic knock-out was performed according to [34, 35].” Considering this is a paper about recombinant strain description, a brief summary of the key steps and any protocol variation would be better than only a citation from another paper.

Line 140: 24 h after transfection… Do not start a sentence with a number. Rephrase to “Twenty-four hours after transfection”

Results and Discussion

Lines 177-181. Do not include these sentences in the results. They are better suited to Material and Methods

Lines 182-184. Please consider typing by using genomic data instead PCR-RFLP because the last has many drawbacks such as subjective interpretation, homoplasy, etc.

Lines 185-186: “TcI is the DTU that circulates most broadly among humans, and is correlated mostly with cardiomyopathic symptoms [40, 41].”  This is not entirely true because TcV is the most common DTU in humans at the Southern cone and it is also associated with cardiomyopathic symptoms. Please rephrase to “TcI is one of the DTUs that…” or something like that.

Lines 197-199. Please consider to remove some data about sequencer (e.g. HiSeq 2500) and sequencing kit (125bp paired end) which are better suited to MM.

Line 197: “67,187,531 reads that passed quality control” How many raw reads did you received?  What was the parameters or criteria of the “quality control”.

Line 202. “The estimated coverage, again assuming a genome size of 53.3 Mb, was 12-fold.” This is wrong. Nanopore has preference for short DNA molecules (when DNA is not fragmented) and it usually favor minicircle sequencing against longer fragments (look at your Figure 2). Consequently, most reads are from minicircles and less than a half are nuclear sequences. In addition, 53.2 Mb is the haploid (not diploid) (nuclear) genome size for Dm28c according to [42]. Consequently, you should calculate the coverage considering non mitochondrial sequences at least for nanopore which has higher bias.

Lines 203-204: It is unclear how did you identified nuclear, maxicircle, minicircle sequences based only on GC content and read length. Don’t you mapped/blasted against a reference? The figure adds a legend “not.mapped”. However, there is no information about mapping against a reference in methodology.

Lines 216-217: “The total assembly amounted to 28.2 Mb in 492 contigs; the nuclear genome had a haploid size of 27.9 Mb in 397 contigs (Table 1).” This is a small genome size compared to other TcI strains. Why? You should discuss this result.

Lines 220-221: “Gene prediction was performed using GLIMMER [43] with the standard codon table. The genome of T. cruzi Dm28c [42] served as training set.” This sentence is better suited to Material and methods. Additional parameters should also be described if they were used.

Lines 261-269: better suited to Material and Methods

Lines 283-286. better suited to Material and Methods

References

Check for species names. They should be in italics

Author Response

We thank the Reviewer for the detailed comments. Based on these comments we have substantially improved the manuscript. Quality control measures, experimental details, etc. were added; please see the enclosed file with all the changes highlighted in yellow.

Line 14: Please use ‘T. cruzi’ instead ‘Trypanosoma cruzi’ when the full species name was already mentioned.

Done.

Line 28: The sentence "Chagas disease is a neglected tropical disease and at the same time also a most elusive disease." can be rephrased to avoid redundancy (3x the word ‘disease’).

The sentence was simplified.

Lines 59-60. “Here we present a new reference strain, T. cruzi STIB980…” Is this strain a new strain? Which are the differences between this strain and the STIB980 presented in reference [21] ( PLoS Negl Trop Dis 2020, 14, e0008487). In addition, this sentence is contradictory to the sentence in line 177 “Trypanosoma cruzi STIB980, originally received in 1983 from Prof. Antonio Osuna, University of Granada…”

We agree that the strain is not really new. The fact that it can serve as a reference strain is new. The sentence was reworded.

Line 104. Which Flow cell did you use?

Information was added.

Line 107. There is no detail about quality control. Which were the criteria? Did you discard low quality reads?

Details on DNA quality control criteria and settings for trimming of reads were added. We also moved the information from Results to Material and Methods as suggested.

Line 108-109. Did you trim Illumina sequences before assembling? Why did you select such Assemblers (please justify). Which parameters did you use? Which results did you obtain from different kmer sizes?

Yes, the Illumina reads were trimmed. The parameters were added, plus more details on kmers.

Lines 110-111. How did you map illumina reads to Nanopore reads to use Pilon? Which software did you use? Which parameters?

BWA MEM was used with default parameters. Information was added.

Lines 133-134. “CRISPR-Cas9 mediated genetic knock-out was performed according to [34, 35].” Considering this is a paper about recombinant strain description, a brief summary of the key steps and any protocol variation would be better than only a citation from another paper.

The key steps had already been summarized. The order of the sentences was changed to clarify this.

Line 140: 24 h after transfection… Do not start a sentence with a number. Rephrase to “Twenty-four hours after transfection”

Done.

Lines 177-181. Do not include these sentences in the results. They are better suited to Material and Methods

The origin of STIB980 was moved to Materials and Methods.

Lines 182-184. Please consider typing by using genomic data instead PCR-RFLP because the last has many drawbacks such as subjective interpretation, homoplasy, etc.

We added genomic data for genotyping, and they confirm that STIB980 is a DTU I strain (new Figure 2).

Lines 185-186: “TcI is the DTU that circulates most broadly among humans, and is correlated mostly with cardiomyopathic symptoms [40, 41].”  This is not entirely true because TcV is the most common DTU in humans at the Southern cone and it is also associated with cardiomyopathic symptoms. Please rephrase to “TcI is one of the DTUs that…” or something like that.

The sentence was corrected.

Lines 197-199. Please consider to remove some data about sequencer (e.g. HiSeq 2500) and sequencing kit (125bp paired end) which are better suited to MM.

Information was transferred to Material and Methods.

Line 197: “67,187,531 reads that passed quality control” How many raw reads did you received?  What was the parameters or criteria of the “quality control”.

The parameters for quality control were added.

Line 202. “The estimated coverage, again assuming a genome size of 53.3 Mb, was 12-fold.” This is wrong. Nanopore has preference for short DNA molecules (when DNA is not fragmented) and it usually favor minicircle sequencing against longer fragments (look at your Figure 2). Consequently, most reads are from minicircles and less than a half are nuclear sequences. In addition, 53.2 Mb is the haploid (not diploid) (nuclear) genome size for Dm28c according to [42]. Consequently, you should calculate the coverage considering non mitochondrial sequences at least for nanopore which has higher bias.

The coverage was re-calculated for total genome and nuclear genome only.

Lines 203-204: It is unclear how did you identified nuclear, maxicircle, minicircle sequences based only on GC content and read length. Don’t you mapped/blasted against a reference? The figure adds a legend “not.mapped”. However, there is no information about mapping against a reference in methodology.

It was actually possible to use GC content combined with length to filter kDNA. The maxicircle DNA was further identified with blastn. This is more evident from the whole contigs than from the reads; new Figure 3B added. 

Lines 216-217: “The total assembly amounted to 28.2 Mb in 492 contigs; the nuclear genome had a haploid size of 27.9 Mb in 397 contigs (Table 1).” This is a small genome size compared to other TcI strains. Why? You should discuss this result.

We do not have an evidence-based explanation for this result. The repetitive nature of the T. cruzi genome may lead to overestimation or underestimation of the genome size. But this is too speculative to discuss. We just refer to a reference that confirms the high variability of T. cruzi genome sizes (new Ref. 52). The reports of the haploid genome size of T. cruzi Dm28c are equivocal; we refer to Berná et al. Microb Genom 2018, 4: e000177 for our calculation.

Lines 220-221: “Gene prediction was performed using GLIMMER [43] with the standard codon table. The genome of T. cruzi Dm28c [42] served as training set.” This sentence is better suited to Material and methods. Additional parameters should also be described if they were used.

The sentence was moved to Material and Methods.

Lines 261-269: better suited to Material and Methods

Information on origin of plasmids was transferred to Material and Methods.

Lines 283-286. better suited to Material and Methods

Information on origin of plasmids was transferred to Material and Methods.

References. Check for species names. They should be in italics.

Done.

Reviewer 3 Report

The manuscript highlights that Trypanosoma cruzi STIB980  strain ability to grow well in culture as axenic epimastigotes or intracellular amastigotes makes it a valuable tool for drug research and studies on host-pathogen interactions. The manuscript also highlights the versatility of T. cruzi STIB980 by mentioning its application in genetic transfection, reporter gene expression, and CRISPR/Cas9-mediated gene editing. This showcases its potential to facilitate various aspects of research related to Chagas disease. Furthermore, the description of the genome sequencing process, which combines Illumina and Oxford Nanopore technologies, adds credibility to the quality of the genetic data obtained. The assembly of a high-quality nuclear haplotype assembly with specific statistics on open-reading frames provides essential details for researchers interested in using this strain.

Overall, the manuscript effectively communicates the significance of Trypanosoma cruzi STIB980 and its potential impact on Chagas disease research. It combines technical details with practical implications, making it a well-structured and informative piece of scientific writing.

Author Response

We thank the Reviewer for the positive comments.

Round 2

Reviewer 2 Report

This is an improved version of the manuscript. It has enough details to ensure reproducibility of the genomic analyses which was the major concern in the previous version.